# Complete Blood Counts and Research Parameters in the Detection of Myelodysplastic Syndromes

**DOI:** 10.3390/diagnostics14131322

**Published:** 2024-06-21

**Authors:** Eloísa Urrechaga, Mónica Fernández, Urko Aguirre

**Affiliations:** 1Laboratory, Galdakao-Usansolo Hospital, 48960 Galdakao, Spain; 2Hematology, Hospital Universitario Araba, 01009 Vitoria-Gasteiz, Spain; monica.fernandezperez@osakidetza.eus; 3Research Unit, Osakidetza Basque Health Service, Barrualde-Galdakao Integrated Health Organisation, Galdakao-Usansolo Hospital, 48960 Galdakao, Spain; urko.aguirrelarracoechea@osakidetza.eus; 4Kronikgune Institute for Health Services Research, 48902 Barakaldo, Spain

**Keywords:** myelodysplastic syndromes, cytopenia, complete blood count, research use only parameters, Mindray BC 6800 Plus analyzer

## Abstract

The diagnosis of Myelodysplastic syndromes (MDS) is frequently challenging, especially in terms of the distinction from the other non-neoplastic causes of cytopenia. Currently, it is based on the presence of peripheral blood cytopenias, peripheral blood and bone marrow dysplasia/blasts, and clonal cytogenetic abnormalities, but MDS diagnostic features are polymorphic and non-specific. We investigated the utility of complete blood count (CBC) and research parameters (RUO) from the analyzer BC 6800 Plus (Mindray Diagnostics) to discriminate MDS-related cytopenia. Methods: 100 samples from healthy individuals were used to establish the values of research parameters in normal subjects. A retrospective study was conducted including 66 patients diagnosed with MDS, 90 cytopenic patients due to other diseases (cancer patients receiving therapy, aplastic anemia, other hematological malignancies) and 50 with macrocytic anemia. The Wilcoxon test was applied to detect statistical differences among the groups of patients, considering *p* < 0.05 significant. The diagnostic performance of the RUO parameters for discriminating MDS among cytopenias was evaluated using receiver operating characteristic (ROC) curve analysis. Amultivariable logistic regression model was performed to identify the potential predictors for having MDS. The area under curve (AUC) and the Hosmer–Lemeshow test of the model were assessed. The performance of the model was verified in a prospective study including 224 cytopenic patients (validation group). Results: In the MDS group, the mean cell volume (MCV), percentage of macrocytic red cells (MAC), red cell distribution width (RDW) and immature platelets fraction (IPF) had increased values compared to the cytopenic and normal patients, while platelets, red and white cell counts, Neu X (related to the cytoplasmic complexity of neutrophils), Neu Y (related to nucleic acid content) and Neu Z (related to cell size) were lower (*p* < 0.001). Neu X, Neu Y, and Neu Z showed higher AUC for detecting MDS > 0.80; MAC, RDW and IPF AUC > 0.76. The multivariable model demonstrated that Neu X and Neu Y could be used in the recognition of MDS, AUC 0.88. In the validation group, 89.0% of the MDS patients were well classified. Conclusion: MDS are common malignant disorders with a poor prognosis, and early diagnosis is warranted for optimal benefit from treatment. RUO gain insights to detect dysplasia of MDS and could be used in the differential diagnosis of MDS from cytopenias of other etiologies.

## 1. Introduction

Myelodysplastic syndromes (MDS) are a group of clonal stem cell disorders characterized by ineffective and dysplastic hematopoiesis, which result in one or more cytopenias (anemia in most cases, as well as thrombocytopenia and/or neutropenia) [1].

The supportive therapy has been replaced with advanced treatment of high risk MDS, but about 30% of MDS cases progress to acute myeloid leukemia. These facts make the early and correct diagnosis of MDS crucial [2]. The risk of developing MDS increases with age and its diagnosis is frequently challenging because the clinical presentation is insidious dependent on the severity and type of cytopenia [3,4].

MDS is suspected upon detection of cytopenia and morphological changes of peripheral blood cells, and the diagnosis/prognosis must be confirmed by bone marrow aspiration, flow cytometry, cytogenetic and molecular analyses [5]. The recommended cytopenia thresholds for defining MDS are hemoglobin (Hb) < 100 g/L, absolute neutrophil count (ANC) < 1.8 × 10^9^/L, and platelet count < 100 × 10^9^/L [6], but it is important to consider that blood counts above these thresholds do not exclude MDS [7].

Hematology analyzers perform complete blood counts (CBC) and the initial classification of blood cells. Due to technological progress over past decades, the number of parameters reported has increased, as well as the so-called research use only parameters (RUO). In addition to the leukocyte differential counts, they can also provide cell population data (CPD) parameters reflecting morphological characteristics of the cells [8]. But dysplasia, including erythroid and megakaryocytic lineage can be observed in MDS.

The Mindray BC-6800 Plus analyzer (Mindray Biomedical Electronics Co., Ltd., Shenzhen, China) reports not only leukocyte CPD but also red blood cell (RBC)-derived parameters, including the percentage of macrocytic RBC (MAC), the percentage of hypochromic or hyperchromic RBC, or immature platelets fraction (IPF).

All those parameters give insights into the morphology of blood cells which could be distinctively disrupted in MDS when compared with cytopenias of a different etiology. In this study, we hypothesized that the novel parameters RUO could be used to detect dysplasia of MDS and might help to identify MDS in a routine clinical setting.

## 2. Patients and Methods

### 2.1. Instrumentation

#### 2.1.1. Mindray Technology

The Mindray BC-6800 Plus analyzer adopts shrated flow impedance combined with fluorescent flow cytometry with a semi-conductor diode laser (SF Cube analysis technology) [9].

The red blood cell (RBC) channel combines the 2 optical signals, side scatter (SS) and forward scatter (FS), to measure subsets of RBCs based on size (micro/normo/macrocytes) and hemoglobin concentration (hypo/normo/hyperchromic). The percentages of each subset are reported and the V/HC cytogram (Mie map) illustrates the distribution (Figure 1). Not only is red cell distribution width (RDW) reported, but hemoglobin distribution width (HDW) is also reported.

Leukocytes (WBC) are counted and classified in different analytical channels, using lysing and fluorescent staining reagents, and channel-specific detectors for FS, SS, and fluorescence of each cell, elaborating information on cell size, cytoplasmic granules, and nuclear characteristics. All measurements are combined in scatterplots, which provide a visual representation of cell cluster distribution according to the analyzed properties. The intensity of fluorescent signal reflects the degree to which the cells are stained and is plotted along the *Y* axis. The internal complexity is plotted along the *X* axis and the cell size is plotted along the Z axis. Cell population data (CPD) are the numerical coordinates to compose those clusters in the scattergram according to the morphology of the leukocyte subsets (Figure 2).

Reticulocyte counts are measured from FS and side fluorescence signals after staining with a proprietary fluorochrome. The optical platelets count can also be obtained from this channel.

Immature platelet fraction (IPF) can be detected and quantified since there is a large amount of RNA in IPF, which produce higher FL (RNA) and FS (size) signals (Figure 3).

Quality control assurance was carried out according to manufacturer’s instructions. The quality of data was validated by routine use of BC-6D and BC-RET quality control materials (Mindray Diagnostics, Shenzhen, China), based on three different levels.

#### 2.1.2. Implantation: Quality Control and Metrological Aspects

Quality control assurance was carried out according to manufacturer’s instructions. The quality of data was validated by routine use of BC-6D and BC-RET quality control materials (Mindray Diagnostics, Shenzhen, China), based on three different levels; mean, standard deviation (SD) and coefficient of variation (CV) were recorded daily for CBC and reticulocytes establish the between-batch precision.

Within run precision can be checked automatically; fresh blood samples (with different concentrations of individual parameters) are processed ten consecutive times; the counter calculates mean, standard deviation (SD) and coefficient of variation (CV).

All blood specimens used in these studies and stability verification were residual material that remained after all medically requested tests had been completed. The patient identification was known only to the laboratory.

CBC parameters may be affected by several preanalytical phase variables which could potentially bias results. These include time between phlebotomy and analysis, and storage/transport conditions.

For the stability study, 25 blood specimens were analyzed, including healthy subjects and samples in a wide range of RBC and WBC values.

The specimens were processed (T0) and left at room temperature (RT).

Analysis was performed after 2, 4, and 8 h of storage.

Then samples were spiked in 2 aliquots, one stored at room temperature and the other refrigerated at 4 °C overnight; all were processed after 24 h.

The stability limits (SL) were calculated SL = ±1.65 Cvi.

Cvi is the between-day imprecision calculated from the internal quality control Mindray BC-RET level 2 for each parameter in the 3 months period in use.

Percentage deviations (PD) were calculated PD = 100/n × Σ (Yi − Xi/Xi).

Xi value at T = 0, Yi results at 2, 4, 8 and 24 h; n number of samples.

PD < SL is the criteria to consider stability.

### 2.2. Patients

The study was retrospective and conducted in the Laboratory of the Galdakao-Usansolo Hospital, after receiving approval from the Research Ethics Committee.

The diagnosis of each patient was retrieved from their laboratory and medical files.

First, a retrospective study was conducted. In total, 100 samples from healthy individuals were used to establish the values of RUO in normal subjects. All blood specimens were residual material that remained after all medically requested tests had been completed.

Data of the 66 patients with MDS were recorded upon diagnosis, which was established according to the 2016 WHO criteria. The integrated diagnostic procedure was based on clinical data and the combined results of CBC, bone marrow examination, enumeration of blast cells and the degree of dysplasia. A marrow karyotype was performed in all cases.

There were a total of 50 patients with macrocytosis, due to lack of vitamin B12 or folate, without other cytopenias nor morphological flags in leukocytes nor platelets. Cytopenia was defined as values below the reference ranges established. There were 90 samples with cytopenia due to other diseases (cancer patients receiving therapy, aplastic anemia, other hematological malignancies).

The performance of the model was verified in a prospective study which included 224 cytopenic patients (validation group), 55 MDS and 169 with non-neoplastic causes of cytopenia.

Peripheral blood smears of all patients recruited in the retrospective and the prospective studies were revised manually by hematologists, and the dysplasia features were confirmed morphologically.

### 2.3. Statistics Analysis

The Kolmogorov–Smirnov test was used to detect whether distributions were normal or skewed. A preliminary exploratory data analysis was performed with medians and interquartile range (IQR). The non-parametric Wilcoxon test was applied to detect statistical differences among groups.

The diagnostic performance of the RUO parameters in the differential diagnosis of MDS versus other cytopenias was evaluated using receiver operating characteristic (ROC) curve analysis.

A multivariable logistic regression model was performed to identify the potential predictors for having MDS. The area under the curve (AUC) value and the test of Hosmer–Lemeshow (calibration) of the final model was assessed. Odds ratios (ORs) were also calculated.

In the validation group, the percentages of correct classification were recorded.

All the statistical procedures were performed using SSPS statistic software 29.0.0. A *p*-value < 0.05 was deemed to be statistically significant.

## 3. Results

The results obtained for the precision study of the RUO parameters are shown in Table 1.

Mean cell volume (MCV), red cell distribution width (RDW), and the percentages of microcytic and macrocytic red cells (MIC, MAC) were stable 8 h after collection, whereas hypochromic red cells (HYPO) showed a mean increase and hyperchromic cells (HYPER) a mean decrease over the SL of 8.7% and 13.5%, respectively, after 4 h.

Storage at 4 °C showed a reduction in osmotic swelling but the variations observed in MIC and MAC were over the SL.

CPD parameters were found to be more stable when maintained at room temperature for up to 2–4 h, and the opposite was found for IPF%. Neu X, Neu Y and Neu Z remained stable for 8 h while, by this time, there was a significant false increase in the IPF% values.

Delayed sample analysis for organizational or technical reasons are not rare in clinical practice. The specimen whole blood is particularly affected by storage temperature and time of incubation: a significant delay and/or poor storage specimens could lead to unreliable values.

Our results suggest that a highly standardized pre analytical process should be implemented for those RBC- and platelet-derived RUO parameters, while leukocyte CPD seem to be more stable.

Table 2 summarizes the analytical data of the recruited patients.

In our group of MDS patients, 29% of them suffered anemia, with no other cytopenia, and 30% of the anemia was macrocytic. Isolated leukopenia affected 4%, and 32% of them were neutropenic. Thrombopenia affected 34% of the patients and was associated with anemia in 54% of the patients and with leukopenia in 37% of them. Three cell lines were affected in 33% of the MDS patients.

MCV, MAC, RDW, HDW, and IPF in the MDS group had higher values compared to the cytopenic patients. When this set of parameters was compared with the macrocytic anemia group, only IPF had higher values. Neu X, Neu Y and Neu Z were significantly lower in the MDS patients (Figure 4, Figure 5, Figure 6, Figure 7 and Figure 8).

The results of the ROC analysis are summarized in Table 3. Neu X, Neu Y and Neu Z showed AUC > 0.80; IPF, MAC and RDW AUC > 0.70.

The results of the multivariable logistic regression model are showed in Table 4 and demonstrate that Neu X and Neu Y could be used in the recognition of MDS among other cytopenias.

In the validation group 81.8% of the MDS and 82.8% of non-MDS cytopenic patients were well classified. The 11 false negative MDS patients suffered mild anemia with no other cytopenia, while the three cellular lineages were affected in the false positive group; 82.1% of the group was well classified.

## 4. Discussion

The diagnostic features of MDS are non-specific and polymorphic, and the standard CBC give little information to help in the detection of the disease. The diagnosis depends on the morphological examination of the peripheral blood and bone marrow [2].

In the present study we have found significant differences in all lineages of MDS-cytopenia, with increased MCV, MAC, RDW and IPF, while the dysplasia in neutrophils is evidenced by Neu X, Neu Y and Neu Z low values. It must be stated that those RUO parameters are privative of the Mindray analyzers; other manufacturers apply different technologies and also report RUO parameters characteristic of the specific counter. These variabilities among analyzers makes it difficult for the harmonization and is a true drawback which makes the universal use of RUO parameters difficult.

The aim of this study was to try to find parameters provided by the Mindray 6800 Plus analyzer that could help routine laboratories to detect MDS. We have found that the structural neutrophil parameters could be useful for this purpose. The multivariable logistic regression model showed that Neu X and Neu Y are the most potent predictors for having MDS. Sun et al. based on the same counter found that Neu X and Neu Z can predict neutrophil dysplasia [10].

The dysplasia of neutrophils in MDS is defined by microscopic findings including nuclear hypolobulation, hypersegmentation, bizarre lobulation, clumping, cytoplasmic hypogranularity and pseudo-Chediak-Higashi granules and small size [11]. Recent studies have found that CPD reflecting the cellular characteristics of neutrophils like cell internal complexity or granularity, can be used to detect dysgranulopoiesis [12].

Despite technical matters, the identification of MDS through CBC and RUO data related to the granulocyte dysplasia has been explored over years. Different researchers have conducted studies using Sysmex analyzers [13,14,15,16,17,18].

Also, predictive scoring systems have been proposed using these parameters for the diagnosis of MDS, even using advanced machine learning algorithms [19,20]. Few data from Beckman Coulter and Abbott counters have been published [21,22,23,24].

Regarding platelets and derived new parameters, our study demonstrates that the immature fraction presents opposite trends in MDS versus macrocytic anemia and non-MDS cytopenia.

Despite the technical matters, similar to those previously stated for RUO parameters, IPF has been considered a reliable marker for the differential diagnosis of thrombocytopenic disorders and can contribute to distinguish peripheral thrombocytopenia with high diagnostic performance [25,26].

Our data are in concordance with previous published data: increased values have been found in patients with MDS, even in the absence of thrombocytopenia; in such cases, IPF would not be associated with an increase in megakaryopoiesis, but the underlying mechanism remains unknown. Moreover, a relationship between the increase in IPF and the poor prognosis in patients with MDS has been suggested [27,28].

Macrocytic anemia, a hallmark of MDS, is due to ineffective erythropoiesis with a high degree of apoptosis of marrow erythroid progenitors. MCV is usually increased in a many non-clonal anemias: liver diseases and endocrinological, vitamin deficiencies, etc. Diagnosis of MDS when anemia is the only abnormality can be complicated [29], especially in the older population, in whom a certain degree of anemia is expected and considered normal by many clinicians. The early diagnosis enables optimal care of these patients, but it is difficult to detect those patients whose CBC only present a mild anemia, as our results show.

Several articles have highlighted the contribution of RBC analysis to MDS diagnosis. Using Sysmex-XN, Boutault et al. showed that MCV was significantly increased in MDS patients versus non-clonal cytopenias and included MCV in a score [19]. Ravalet et al. reported similar MVC using the DxH 800 Beckman Coulter analyzer [23].

In microscopic examinations of RBCs from MDS patients, anisocytosis and poikilocytosis are also frequently observed, which can be quantified by means of RDW. Certain analyzers can report other distribution parameter similar to RDW: HDW or Hb distribution width, which reflects anisochromia, and the distribution of Hb in the individual red cells. Both parameters could be useful in the screening of MDS among other anemias, as suggested by Hwang et al., using the Abbott Alinity-hq analyzer. RDW, HDW, MCV and MCH were higher in MDS patients than in non-clonal cytopenias [24], and the values are in full agreement with our results.

Among RUO parameters, the best attention has been paid to those related to the granulocyte dysplasia, which is typical feature of MDS. Despite different technologies, Neu X provided by analyzers from different brands was correlated with the morphological change of neutrophil-decreased granules. Granularity index can be derived from data of the Mindray [10] Sysmex, Abbott and Beckman Coulter analyzers [30,31,32].

Neu X, Neu Y and Neu Z can reflect in numbers the severe alterations of the neutrophil morphology, internal complexity (Neu X), nucleic acid content (Neu Y) and cellular size (Neu Z). In our study, all of them presented significant lower values compared to the values seen in other maladies, macrocytic anemia and cytopenic non-MDS patients.

Our results confirm those previously reported using Sysmex analyzers (those using fluorescence flow cytometry like the analytical principles on Mindray counters). NEUT-X and NEUT-Y are privative of Sysmex counters, with similar morphological meaning as Neu X and Neu Y, respectively.

Le Roux et al., using Sysmex XE-2100 counter concluded that a low NEUT-X value in combination with anemia was strongly correlated with MDS [30]. Furundarena et al. with the same analyzer verified these results and suggested the value of NEUT-Y to detect neutrophil dysplasia arising from MDS [13].

It must be stated that, due to the lack of standardization among different brands, CPD cannot be used interchangeably, and instrument-specific clinical decision limits are required. This study was conducted in one hospital, and thus the cut offs selected must be verified in a multicentric evaluation.

The use of Neu X and Neu Y are convenient and objective markers obtained from the automated analyzer along with the CBC. This could help to detect MDS rapidly without additional costs, improving the efficiency of the laboratory process and contributing to making final diagnose of the patients in time.

## 5. Conclusions

The diagnosis of MDS is frequently challenging, especially in terms of the distinction from the other non-neoplastic causes of cytopenia. Research parameters from hematology analyzers may be useful to discriminate MDS-related cytopenia. Parameters such as Neu X (related to the cytoplasmic complexity) and Neu Y (related to nucleic acid content) show promise to detect dysplasia of MDS and aid to recognize MDS from cytopenias of other etiologies.

## Figures and Tables

**Figure 1 diagnostics-14-01322-f001:**
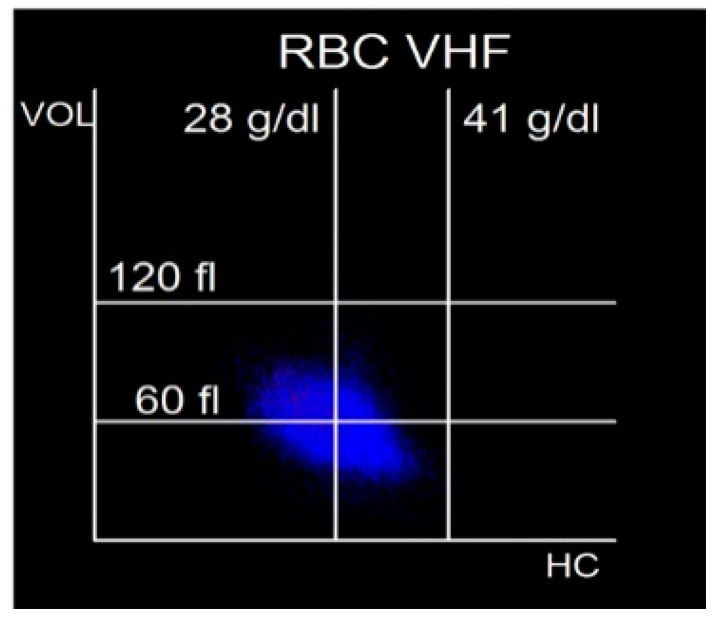
The V/HC cytogram, Mie map, and Hb concentration is plotted along the *X* axis and cell volume is plotted along the *Y* axis. Only red blood cells appear on this cytogram. Markers organize the cytogram into 9 distinct areas of red blood cell morphology. On the *X* axis, Hb concentration markers are set at 28.0 g/dL and 41.0 g/dL. RBC with a Hb concentration < 28.0 g/dL are hypochromic, while cells with a Hb concentration > 41.0 g/dL are hyperchromic. On the *Y* axis, red cell volume markers are set at 60 fL and 120 fL. Red cells with a volume < 60.0 fL are microcytic, while cells with a volume > 120.0 fL are macrocytic.

**Figure 2 diagnostics-14-01322-f002:**
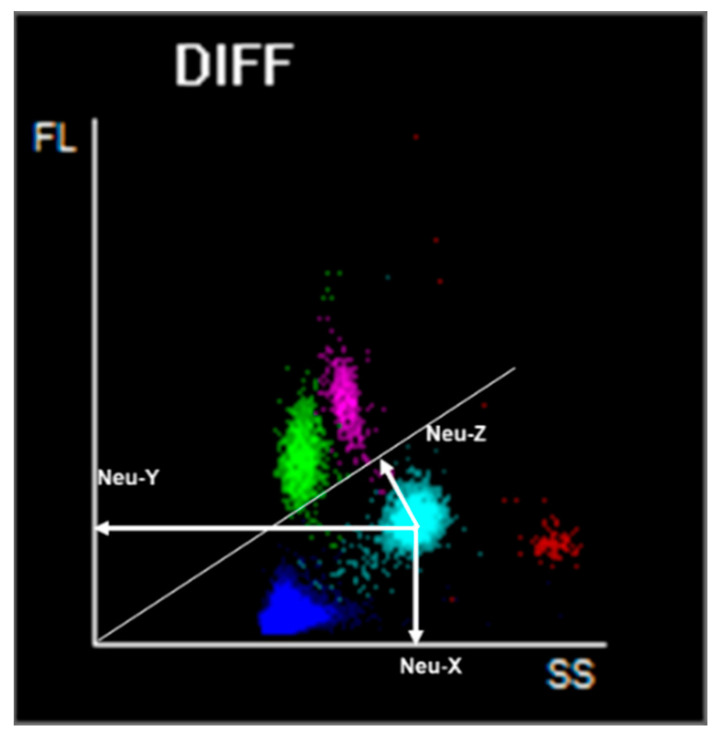
Each leukocyte can be described by 3 optical signals appearing in the DIFF scattergram: the coordinates to compose this scattergram are the morphometric parameters, so called Cell Population Data. CPD are related to the morphology and functional characteristics of the leukocyte populations and report on internal cellular complexity: Neu X, Lym X, Mon X; nucleic acid content: Neu Y, Lym Y, Mon Y; and cell size: Neu Z, Lym Z, Mon Z.

**Figure 3 diagnostics-14-01322-f003:**
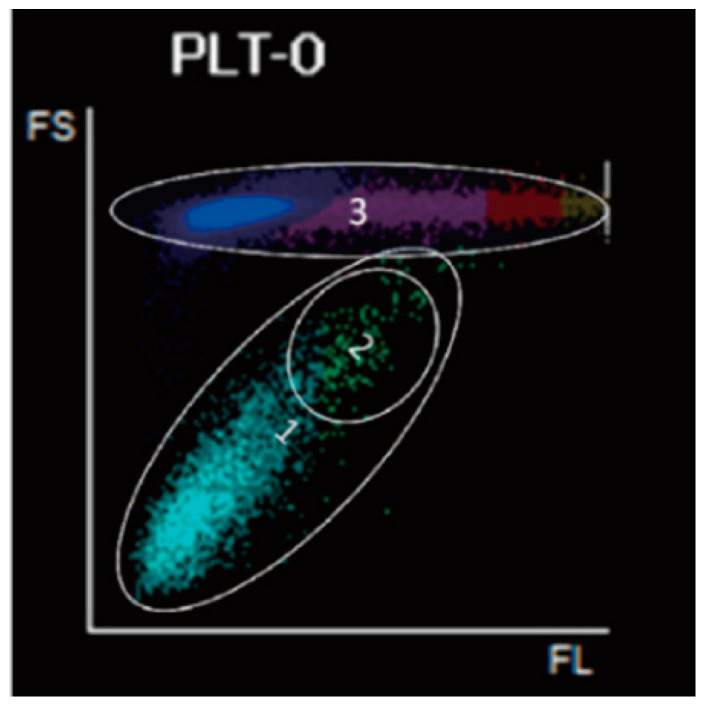
PLT-O Scattergram (reticulocyte channel). *X* axis represents the fluorescence signal; *Y* axis represents forward scatter light. IPF is gated within a certain area showing stronger fluorescent signals. PLT-O optical platelets. 1: Platelets. 2: Immature platelets. 3: Optical red blood cells.

**Figure 4 diagnostics-14-01322-f004:**
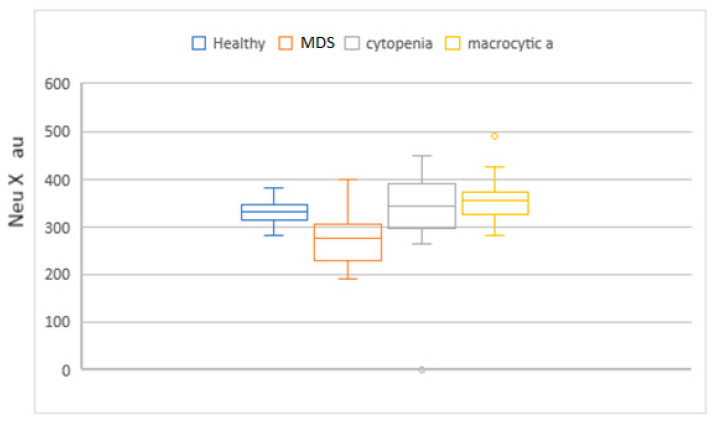
Box and whisker plots of the NEUT-X parameter for the different patient groups. Median in the MDS group was significantly lower (279 arbitrary units, au) when compared to the groups, macrocytic anemia and no-MDS cytopenias (355 au and 350, respectively).

**Figure 5 diagnostics-14-01322-f005:**
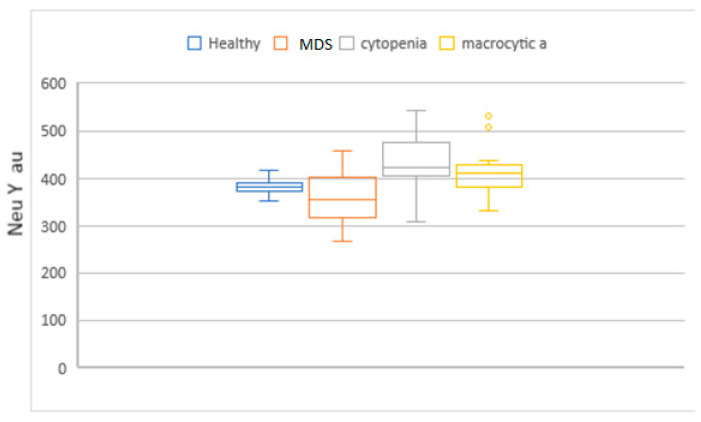
Box and whisker plots of the NEUT-Y parameter for the different patient groups. Median in the MDS group was significantly lower (362 arbitrary units, au) when compared to the groups, macrocytic anemia and no-MDS cytopenias (413 au and 436 au, respectively).

**Figure 6 diagnostics-14-01322-f006:**
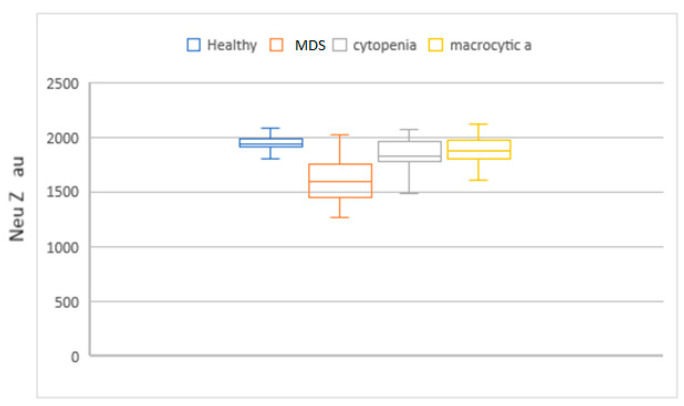
Box and whisker plots of the NEUT-Z parameter for the different patient groups. Median in the MDS group was significantly lower (1604 arbitrary units) when compared to the groups, macrocytic anemia and no-MDS cytopenias (1869 au and 1849 au, respectively).

**Figure 7 diagnostics-14-01322-f007:**
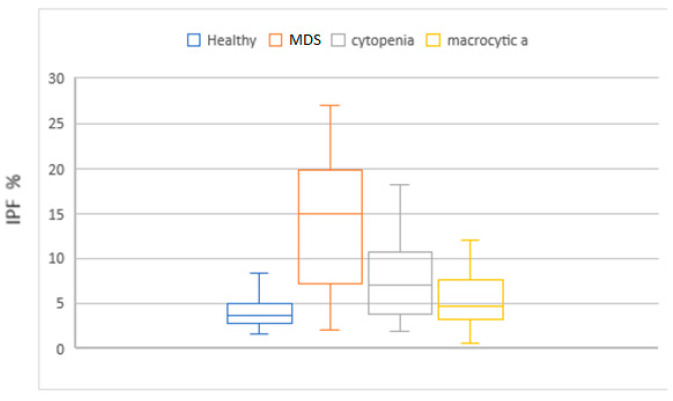
Box and whisker plots of the immature platelet fraction for the different patient groups. Median in the MDS group was significantly higher (14.4%) when compared to the groups, macrocytic anemia and no-MDS cytopenias (5.4% and 7.5%, respectively).

**Figure 8 diagnostics-14-01322-f008:**
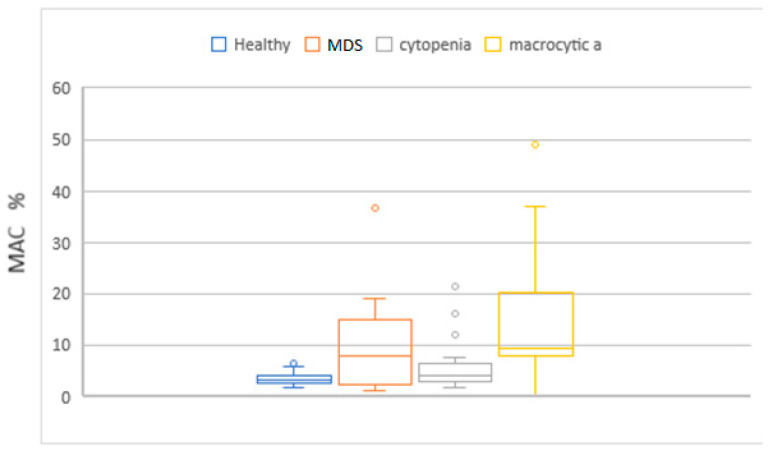
Box and whisker plots of the percentage of macrocytic red cells for the different patient groups. Median in the MDS group was significantly higher (9.8%) when compared to the no-MDS cytopenias group (5.8%).

**Table 1 diagnostics-14-01322-t001:** To quantify within-run precision, 10 peripheral blood samples were processed 10 consecutive times; the counter can calculate mean, standard deviation (SD) and coefficient of variation (CV). We report the results obtained for RUO parameters.

	Mean	SD	CV	Mean	SD	CV	Mean	SD	CV
Neu X	322	1.1	0.35	375	2.7	0.75	367	3.8	1.0
Neu Y	458	3.9	0.85	393	2.8	0.72	392	1.9	0.48
Neu Z	1931	6.2	0.32	1963	7.5	0.39	1806	11.1	0.61
Lym X	94	1.97	2.0	83	0.77	0.92	87	1.2	1.37
Lym Y	596	8.5	1.43	611	8.5	1.39	578	6.6	1.14
Lym Z	979	10.1	1.1	971	4.4	0.45	957	8.8	0.92
Mon X	214	4.5	1.9	185	2.5	1.35	201	1.9	0.95
Mon Y	869	20	2.3	904	13.8	1.53	843	10.7	1.27
Mon Z	1364	18.7	1.33	1333	14.7	1.1	1351	9.9	0.73
IPF	7.3	0.54	7.4	2.5	0.16	6.4	3.75	0.2	5.3
MIC	0.63	0.06	10.5	0.65	0.05	8.1	21.8	0.19	0.88
MAC	6.9	0.15	2.1	2.7	0.11	4.2	1.21	0.08	7.1
HPER	0.52	0.04	8.07	1.33	0.04	3.6	1.78	0.78	4.4
HPO	0.86	0.08	9.8	0.13	0.04	3.5	3.2	0.09	2.9
RDW	33.8	0.38	1.12	39.8	0.19	0.48	52.9	0.23	0.44
HDW	23.04	0.21	0.94	20	0.29	1.45	40.4	0.33	0.82

Neu X, neutrophils complexity; Neu Y, neutrophils fluorescence; Neu Z, neutrophils size; Lym X, lymphocytes complexity; Lym Y, lymphocytes fluorescence; Lym Z, lymphocytes size; Mon X, monocytes complexity; Mon Y, monocytes fluorescence; Mon Z, monocytes size. Cell Population Data rare reported in arbitrary optical units. IPF immature platelet fraction (%); MIC, microcytic red cells (%); MAC macrocytic RBC (%); HYPER, hyperchromic RBCs (%); HYPO, hypochromic RBCs (%); RDW, red cell distribution width; and HDW, hemoglobin distribution width (g/L).

**Table 2 diagnostics-14-01322-t002:** Complete blood counts and research parameters obtained using the Mindray BC 6800 Plus analyzer in various groups: healthy subjects, macrocytic anemia, cytopenias and myelodysplastic syndromes (MDS). *p* values, significance of the differences when compared with the MDS values.

	Healthy		Macrocytic			Cytopenia			MDS	
	Median	IQR	Median	IQR	*p*-Value	Median	IQR	*p*-Value	Median	IQR
WBC	6.57	4.93–8.77	9.00	4.5–11.7	0.0092	3.25	2.38–4.01	0.018	3.98	2.9–4.95
RBC	4.65	4.22–5.22	3.28	3.08–4.02	0.4678	3.74	3.24–4.15	0.03	3.47	2.99–3.88
Hb	134	125–155	103	79–111	0.3103	102	99–128	0.067	99	96–119
MCV	92.3	87.0–95.5	103.4	99.8–113.8	0.0004	93.7	86.1–100.0	0.001	97.8	90.2–105.1
MCH	28.9	28.5–31.3	31.9	30.5–35.1	0.0049	29.9	28.7–30.3	0.188	28.9	27.5–29.8
MCHC	312	310–322	306	295–318	1	318	314–333	0.178	316	313–330
RDW	44.2	41.0–48.1	52.5	49.1–63.5	0.2227	48.2	42.3–53.7		54.7	48.9–59.4
Platelets	266	190–352	253	148–375	0.0001	85	32–148	0.001	119	62–191
IPF	4.1	2.0–6.4	5.4	2.9–9.8	<0.0001	7.5	3.6–13.2	0.0001	14.4	4.0–23.8
MIC	0.78	0.3–1.8	1.14	0.3–1.5	0.0045	1.15	0.99–4.1	0.04	1.55	1.0–3.2
MAC	3.4	2.3–4.8	10.9	7.8–21.2	0.028	5.8	3–7.8	0.001	9.8	3.1–18
HYPER	1.7	0.8–3.6	2.9	0.7–4.2	0.1301	0.94	0.45–1.6	0.02	2.03	0.7–3.5
HYPO	0.1	0–0.2	0.6	0.2–1.7	0.0169	0.87	0.2–1.5	0.08	0.79	0.1–1.1
HDW	19.1	15.9–23.8	23.9	16.2–38	0.08	22.9	19.1–26.9	0.03	25.2	18.1–29.0
Neu X	332	303–369	355	317–374	<0.0001	350	290–388	0.0001	279	195–305
Neu Y	382	3663–407	413	362–433	0.0009	436	405–475	<0.0001	362	299–403
Neu Z	1951	1873–2061	1869	1702–1987	<0.0001	1849	1780–1981	<0.0001	1604	1378–1771
Lym X	90	86–94	90	84–97	0.278	95	89–100	0.1153	92	86–96
Lym Y	592	565–624	594	569–626	0.0005	634	600–659	0.7947	640	578–675
Lym Z	1021	1002–1042	1011	966–1033	0.5938	1031	999–1057	0.2119	1019	933–1051
Mon X	197	190–208	208	196–222	0.7304	208	191–222	0.7389	206	188–226
Mon Y	858	827–906	881	834–941	0.0025	940	855–979	0.3556	968	865–1039
Mon Z	1387	1339–1451	1413	1354–1441	0.3447	1401	1332–1445	0.8932	1398	1338–1420
Neut	3.62	2.38–5.69	5.95	1.46–7.02	<0.0001	2.22	1.45–3.03	0.05	1.83	1.45–3.39
Lymph	2.27	1.45–3.14	1.99	0.83–2.4	0.04	1.59	1.18–2.22	0.212	1.51	0.99–2.18
Mono	0.44	0.31–0.64	0.9	0.23–0.99	0.001	0.52	0.38–0.73	0.03	0.39	0.19–0.68

Abbreviations: MDS, myelodysplastic syndrome, IQ interquartils; WBC, leukocyte count (×10^9^/L); RBC, red blood cells (×10^12^/L); Hb, Hemoglobin (g/L); MCV, mean cell volume (fL); MCH, mean cell hemoglobin (pg); MCHC, mean cell hemoglobin concentration (g/L); RDW, red cell distribution width (fL); IPF immature platelet fraction (%); MIC, microcytic red cells (%); MAC macrocytic RBC (%); HYPER, hyperchromic RBCs (%); HYPO, hypochromic RBCs (%); HDW, hemoglobin distribution width (g/L); Neu X, neutrophils complexity; Neu Y, neutrophils fluorescence; Neu Z, neutrophils size; Lym X, lymphocytes complexity; Lym Y, lymphocytes fluorescence; Lym Z, lymphocytes size; Mon X, monocytes complexity; Mon Y, monocytes fluorescence; Mon Z, monocytes size; Neut, neutrophils (×10^9^/L); Lymph, lymphocytes (×10^9^/L); Mono, monocytes (×10^9^/L). Cell Population Data rare reported in arbitrary optical units. IQR: Interquartile range.

**Table 3 diagnostics-14-01322-t003:** The performance of the research use only parameters to detect myelodysplastic syndromes among other cytopenias was studied using receiver operating characteristic curves.

	AUC	95% CI	Cut off	Sensitivity %	Specificity %
Neu X	0.836	0.728–0.944	<330 au	80.5	81.3
Neu Y	0.825	0.705–0.944	<375 au	58.9	96.0
Neu Z	0.841	0.731–0.951	<1700 au	83.1	80.8
IPF	0.787	0.642–0.933	>10.5%	72.1	80.9
MAC	0.798	0.714–0.878	>9.8%	50.8	86.1
RDW	0.765	0.677–0.855	>55 fL	70.8	79.2

AUC, area under curve; CI, confidence interval; Neu X, neutrophils complexity; Neu Y, neutrophils fluorescence; and Neu Z, neutrophils size, all reported in arbitrary units. IPF, Immature platelet fraction; MAC, percentage of macrocytic red cells; and RDW, red cell distribution width.

**Table 4 diagnostics-14-01322-t004:** Results of the multivariate logistic regression analysis.

	OR	95%CI	*p* Value	Cut off
Neu X	19.23	2.00–185.26	0.01	<330 au
Neu Y	39.65	2.91–538.99	0.006	<375 au
AUC (95% CI)	0.88 (0.79–0.95)			

AUC, area under curve; CI, confidence interval; OR, odds ratio; Neu X, neutrophils complexity; and Neu Y, neutrophils fluorescence.

## Data Availability

The data collected and analyzed during this study are available from the corresponding author upon reasonable request.

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
