# Peer review of "Complete Blood Counts and Research Parameters in the Detection of Myelodysplastic Syndromes"

_diagnostics, 2024, doi:10.3390/diagnostics14131322_

Round 1

Reviewer 1 Report

Comments and Suggestions for Authors

Review of 

Complete Blood Counts and Research Parameters in the 2

Detection of Myelodysplastic Syndromes

ts important issue to use simple available investigation to diagnose challenging disease specially in developing counteries where advance investigation not always available 

No 

Item 

Request 

1

Health sample is 100 

Is it randomize selection or not 

2

50 sample with macrocytosis 

Is it randomize selection 

3

Retrospective study 

It is retrospective study or case control study 

Table no 1 

It need more suitable formatting 

5

Figures 4-5 

These figures need to more details regarding illustrating of results 

6

Table no 3

Compare MDS with other cytopenia ( who is cytopenia ) what about other groups 

Table no 2 

This table contain two p value ( it need to be more clear) 

No

Item

Response

1.

What is the main question addressed by the research?

How can physicn use simple available investigation to help in diagnosis challenging disease

2.

What parts do you consider original or relevant for the field? What 
specific gap in the field does the paper address? 

Get benefit from available investigation

The gap is, methodology need to be more addressed

3.

What does it add to the subject area compared with other published  material?

Give simple valuble approach for diagnosis

4

What specific improvements should the authors consider regarding the 
methodology? What further controls should be considered?
5posed were addressed and by which specific experiments.

Design of the study, need to re evaluted, randamize of patient and control should be in clear way

5

Please describe how the conclusions are or are not consistent with the evidence and arguments presented. Please also indicate if all main questions 

Conclusion is consist with

6

Are the references appropriate?

Yes

7

Please include any additional comments on the tables and figures and 
quality of the data.

Tables  need minor revission by authors,

Figuers need to discusss in more detials regarding its content

Author Response

1 100 healthy subjects, the selection was randomized, from our daily workload.

2 50 samples with macrocytosis, the selection was randomized, from our daily workload ; all of them had a manual microscopic review of PB slides.

3 The study was retrospective, all patients recruited had a manual microscopic review of PB slides.

4.Table 1  we have changed the table 1

5 Figures, detailed information of the B&W plot has been added

6 Table 3. Cytopenia was defined as values bellow the reference ranges established. Those patients with no-MDS cytopenia suffered  cancer receiving therapy, aplastic anemia, other hematological malignancies.    Healthy subjects were not included , because normal CBC could no trigger any flag

7 Table 2. In this table P values reported are the  significance of the differences of macrocytic anemia group and  cytopenias-no MDS when compared with the MDS values. 

Reviewer 2 Report

Comments and Suggestions for Authors

The Authors provide an interesting paper about CBC values and MDS diagnosis.

1) Why is not cited the new terminology of Myelodisplastic neoplasms (as the WHO 2022?)

2) Were the sample assessed for morphological dysplasia, as the gold standard? In the Discussion, NEU-X, Y and Z are said to evidence dysplasia, was this confirmed morphologically?

Comments on the Quality of English Language

In line 55, comforted should read "confirmed"

Figure 4,5,6,7 should read "MDS", not MSD

Author Response

Reviewer 2

Thank you for your comments to improve our manuscript

1) Why is not cited the new terminology of Myelodisplastic neoplasms (as the WHO 2022?)

Thank you for your comment; we were referring to MDS at diagnosis previously to final classification. We don´t have enough patients to divide MDS in subclasses and get solid results, we must continue the study to compare the patients according to the 2022 classification

2) Were the sample assessed for morphological dysplasia, as the gold standard? In the Discussion, NEU-X, Y and Z are said to evidence dysplasia, was this confirmed morphologically?

Yes , dysplasia was confirmed , in the initial study all the PB slides were revised by the hematologist.

  line 55, comforted should read "confirmed"    Typos are corrected

Figure 4,5,6,7 should read "MDS", not MSD       Typos are corrected

Reviewer 3 Report

Comments and Suggestions for Authors

This a very well written and very interesting study. The data is solid and findings and their relevance are very briefly ana effectively summarized.

some issues:

-in table 2: p-values of MDS group are not visible, was this deliberate?

- introduction: it is important to know which cytopenias are MDS and which are not and which we should closely monitor and which not and which we should biopsy without observation; rather than earyl diagnosis and treatment. We need non-invasive,more objective and better predictors for MDS.

- Again in the introduction, IPF and RBC parameters are of values, beacuse myeloid lineage may not haveen affected ealy on and WBC morphological findings may be unchanged on peripheral smears, and there isn't much to detect about platelet and RBC morphology on peripheral smears.

- in figues you mis-spelt MSD instead of MDS.

- a composite model of non-chemo/non-alcohol/normal folate-B12-copper/persistent cytopenia plus these parameters can be worked out in future study with the aim of higher sensititvity. 

- the timing issue has been poorly discussed, as in the results , you clearly state that IPF is unreliable even as early as after 4h of collection; whereas WBC parameters are more time-independent. This deseves mentioning.

- It was disappointing, IPF and RBC parameters not being significant on MVA logistic regression. Why is that? discussion?

-those false negative MDS patients with mild low Hb levels: were their MDS diagnosis based on solid grounds? 

- does NEU-X and NEU-Y also perfrom well in subclasses of MDS, like thrombocytopenia-only, anemia-only (those without leucopenia)?

Author Response

Reviewer 3

Thank you for your comments to improve our manuscript

-in table 2: p-values of MDS group are not visible, was this deliberate?

Table 2 displays Complete blood counts and research  parameters  in healthy subjects, macrocytic anemia, cytopenias and MDS. P values are the  significance of the differences when healthy subjects, macrocytic anemia, cytopenias groups were compared with the MDS values.

- introduction: it is important to know which cytopenias are MDS and which are not and which we should closely monitor and which not and which we should biopsy without observation; rather than earyl diagnosis and treatment. We need non-invasive,more objective and better predictors for MDS.

Absolute agree, our purpose is to aid in the selection of patients deserving  further studies to get the final diagnosis

- Again in the introduction, IPF and RBC parameters are of values, beacuse myeloid lineage may not have affected ealy on and WBC morphological findings may be unchanged on peripheral smears, and there isn't much to detect about platelet and RBC morphology on peripheral smears.

Absolutely right, we introduced the 3 lineages research parameters also  to make a description of all these values at diagnosis of MDS , because, to the best of our knowledge, no data are  published for Mindray counters and we wanted to compared with data available for Sysmex and Coulter analyzers. Anyway,  the multivariate logistic regression shows the reviewer is complete right.

- in figues you mis-spelt MSD instead of MDS.   Typo has been corrected

- a composite model of non-chemo/non-alcohol/normal folate-B12-copper/persistent cytopenia plus these parameters can be worked out in future study with the aim of higher sensititvity. 

Thank you for this interesting suggestion; unfortunately,  we have very few patients with anemia related Cu deficiency; we are trying to compose a multicentric study focusing to vitamin and trace elements deficiencies.

- the timing issue has been poorly discussed, as in the results , you clearly state that IPF is unreliable even as early as after 4h of collection; whereas WBC parameters are more time-independent. This deserves mentioning.

Thank you  for this suggestion. We have introduced a paragraph to comment this findings

- It was disappointing, IPF and RBC parameters not being significant on MVA logistic regression. Why is that? discussion?

We agree that the final results of the logistic regression were in some way disappointing, but  are in agreement with literature based on data from other counters than Mindray: CPD related to dysplasia of neutrophils are promising markers of MDS.

-those false negative MDS patients with mild low Hb levels: were their MDS diagnosis based on solid grounds? 

We  rely on the medical records and enrolled MDS patients; the false negative MDS in our study were those misdiagnosed only based on our algorithm; all of them were diagnosed MDS patients based on standard procedures

- does NEU-X and NEU-Y also perfrom well in subclasses of MDS, like thrombocytopenia-only, anemia-only (those without leucopenia)?

We don´t have enough patients to divide MDS in subclasses and get solid results. We can continue the study recruiting more patients. 

Round 2

Reviewer 2 Report

Comments and Suggestions for Authors

Given the Authors' reply I would suggest to put in the "Methods" section explicitely that all dysplasia features were confirmed morphologically.

Author Response

Given the Authors' reply I would suggest to put in the "Methods" section explicitely that all dysplasia features were confirmed morphologically.

Thank you for the suggestion, we have included a paragraph with this statement